

# Ecophylogeny of the endospheric root fungal microbiome of co-occurring *Agrostis stolonifera*

Amandine Lê Van[1], Achim Quaiser[1], Marie Duhamel[1,2],
Sophie Michon-Coudouel[3], Alexis Dufresne[1] and Philippe Vandenkoornhuyse[1]

[1] CNRS, UMR6553 Ecobio, Université de Rennes 1, Rennes, France
[2] Department of Biology, Stanford University, Stanford, CA, United States of America
[3] CNRS, UMS3343 OSUR, Université de Rennes 1, Rennes, France

## ABSTRACT

**Background.** Within the root endosphere, fungi are known to be important for plant nutrition and resistance to stresses. However, description and understanding of the rules governing community assembly in the fungal fraction of the plant microbiome remains scarce.

**Methods.** We used an innovative DNA- and RNA-based analysis of co-extracted nucleic acids to reveal the complexity of the fungal community colonizing the roots of an *Agrostis stolonifera* population. The normalized RNA/DNA ratio, designated the 'mean expression ratio', was used as a functional trait proxy. The link between this trait and phylogenetic relatedness was measured using the Blomberg's $K$ statistic.

**Results.** Fungal communities were highly diverse. Only ∼1.5% of the 635 OTUs detected were shared by all individuals, however these accounted for 33% of the sequence number. The endophytic fungal communities in plant roots exhibit phylogenetic clustering that can be explained by a plant host effect acting as environmental filter. The '*mean expression ratio*' displayed significant but divergent phylogenetic signals between fungal phyla.

**Discussion.** These results suggest that environmental filtering by the host plant favours the co-existence of related and similar OTUs within the Basidiomycota community assembly, whereas the Ascomycota and Glomeromycota communities seem to be impacted by competitive interactions which promote the co-existence of phylogenetically related but ecologically dissimilar OTUs.

Corresponding author
Philippe Vandenkoornhuyse,
philippe.vandenkoornhuyse@univ-rennes1.fr

## INTRODUCTION

Plants are colonized by a wide collection of microorganisms forming the plant microbiome. The plant microbiome supports additive functions involved in the plant's adjustments to environmental conditions (for review, *Vandenkoornhuyse et al., 2015*), and thus controls, in part, plant fitness. Understanding the composition and rules of assembly within the plant microbiome is currently a key question in ecology.

Plant roots can be considered as a well-delimited ecological compartment with the endosphere constituting a 'restricted area' (Vandenkoornhuyse et al., 2015). Microbial communities within this compartment differ markedly in their composition from both the surrounding rhizospheric and soil microbial communities (e.g., Bulgarelli et al., 2012; Schlaeppi et al., 2014; Fonseca-García et al., 2016). Beside the well-known mycorrhizal fungi, nitrogen-fixing bacteria, and plant-growth-promoting bacteria (PGPB) the plant microbiome includes a high diversity of associated microorganisms (e.g., Vandenkoornhuyse et al., 2002a; Lambais et al., 2006; Lundberg et al., 2012; Bulgarelli et al., 2012; Peiffer et al., 2013; Schlaeppi et al., 2014). The coexistence of numerous microbial taxa in plant roots is striking and raises the question of the assembly rules underlying these complex communities. However, this aspect remains poorly understood.

To date, most microbial ecology studies have addressed species diversity without taking into account the phylogenetic relatedness among microorganisms. However, our understanding of community assembly can be significantly improved by studying the community's phylogenetic structure (Webb et al., 2002). Assuming that related species share more similar traits than distant species, an analysis of phylogenetic structure can be used to link phylogenetic patterns to ecological processes (Cavender-Bares et al., 2009). According to the niche-based theory, specific phylogenetic patterns can be generated both by inter-specific interactions and environmental filtering by the host plant (e.g., Helmus et al., 2007; Cavender-Bares et al., 2009). In the phylogenetic pattern of overdispersion, species are less related to each other than species assembled at random from a regional pool. This pattern can be promoted, for example, by competition among related species which is expected to limit similarity (Diamond, 1975; Cahill et al., 2008). Similarly, facilitative interactions are known to increase phylogenetic diversity in plant communities when facilitation occurs among distantly related species and favours species overdispersion (Valiente-Banuet & Verdú, 2007). These mechanisms are counterbalanced by environmental filtering that selects for similar traits (Mayfield & Levine, 2010) and may promote phylogenetic clustering. In a clustered phylogenetic pattern, species within the community are more related to each other than expected. Conversely, an absence of phylogenetic structure indicates that the species within the community are a random assemblage of the regional species pool. To be able to draw inferences from phylogenetic structure, the main assumption that phylogenetic relatedness is linked to ecological similarity, must be respected (Kembel, 2009). This assumption is satisfied when a phylogenetic signal can be measured. The phylogenetic signal is a metric used to measure this link by comparing trait similarity to phylogenetic relatedness (Blomberg, Garland & Ives, 2003).

Plants are sessile organisms that have to cope with the environmental changes they experience. One recent idea is that microorganism recruitment within the plant microbiome allows these changes to be buffered (Vandenkoornhuyse et al., 2015). Some filtering by the host plant can therefore be expected and should leave a specific signature in the phylogenetic structure. In this study, the fungal microbiome was defined by analyzing 18S rRNA amplicon sequences produced from co-extracted DNA and RNA from plant root samples to identify the metabolically active microbial community (Klein et al., 2016). These co-extracted DNA- and RNA-based data were used to compare the endospheric

fungal community composition and diversity and the core microbiome amongst 19 co-occurring *Agrostis stolonifera* plants. We introduced the '*mean expression ratio*' as a proxy for functional trait, a functional trait being a measure related to species (i.e., Operational Taxonomic Units, OTUs hereafter) in ecological terms (i.e., activity, interactions) (*Diaz & Cabido, 2001*). In microbial communities these traits are for example, the ability to colonize plant roots, the metabolic activity at different temperatures, or the ability to fix nitrogen. The '*mean expression ratio*' allows each OTU to be described by integrating the information produced from either the DNA or the RNA (i.e., ratio of observed number of sequences for each OTU). Under the hypothesis that traits favourable to an endophytic life style are phylogenetically conserved (*Martiny, Treseder & Pusch, 2013*), we could expect that the '*mean expression ratio*' of related microbial OTUs in plants would be more similar than that of phylogenetically distant OTUs. We investigated these hypotheses by (i) analyzing the phylogenetic structure of 19 endophytic fungal communities from *Agrostis stolonifera* roots (ii) searching for a phylogenetic signal using the '*mean expression ratio*' and, (iii) testing whether the phylogenetic signal was conserved among the main fungal phyla. We limited possible stochastic effects by using spatially aggregated host plants.

## MATERIALS AND METHODS

### Plant harvesting and root sampling

Turf samples (L44 × W34 × D23 cm) were collected from a peatland in the Parc Naturel Régional des Marais du Cotentin et du Bessin, Normandy, France (49.284656–1.393090). The sampled turf vegetation was dominated by *Agrostis stolonifera* (∼94% coverage), a generalist plant often found in disturbed habitats, wetland margins and fields. The other co-occurring plants were *Potentilla anserina*, *Mentha* spp. and *Hydrocotyle vulgaris*. Turfs were placed in a growth chamber for four months at 15 °C with 16 h light per day, and 80% humidity to avoid stochastic environmental effects. Nineteen co-occurring *Agrostis stolonifera* plants were manually harvested from a single turf. During sampling, the turf was divided into 3 disconnected blocks but as there was no block effect in the observed variances, all data were pooled for the analyses. The sampled plants were of similar age with comparable root systems. No difference in the phenotypic traits of sampled plants was noticed. The roots were washed with tap water to remove soil and the surface roots were sterilized by washing three times with 0.1% Triton X-100 and rinsed five times with sterile distilled water (*Vandenkoornhuyse et al., 2007*). The roots were then transferred into RNAlater (Sigma-Aldrich) and stored at −20 °C.

### Nucleic acid extraction and 18S rRNA amplicon sequencing

These sampled roots were crushed into powder with a mortar and pestle in liquid nitrogen. RNA and DNA were co-extracted from the nineteen samples using a modified RNeasy Plant Mini kit (Qiagen, Hilden, Germany) protocol. The on-column DNase digestion was skipped to keep the DNA in the total extracts. For RNA extraction half of the total nucleic extracts were treated with RQ1 DNAse (Promega, Madison, WI, USA) and the complete elimination of DNA was confirmed by 18S rRNA targeting PCR. RNA and DNA quality was checked using the RNA 6000 Pico kit or the High sensitivity DNA kit,

respectively, on a 2100 Bioanalyzer (Agilent, Santa Clara, CA, USA). PCRs were performed using the primer pair SSU0817 (5′-TTAGCATGGAATAATRRAATAGGA-3′) -NS22B (5′-AATTAAGCAGACAAATCACT-3′) targeting a region of about 530 bp of the 18S rRNA gene that includes the variable regions V4 (partial) and V5 (*Borneman & Hartin, 2000*). These particular primers were chosen among a variety of candidate primers after an *in silico* analysis, using Primer Search (*Rice, Longden & Bleasby, 2000*). The chosen set could amplify 94% of the available fungal sequences *in silico*, with the exception of *Microsporidia,* and only 1.3% of *Viridiplantae* in the Silva database 115 (*Quast et al., 2013*). PCR amplifications were performed with fusion primers containing sequencing adaptors, multiplex identifiers (MID) and PCR primers (Table S1). The DNA was amplified by performing direct PCRs on the total nucleic acid extracts using illustra puReTaq Ready-To-Go PCR Beads (GE Healthcare). Two microliters of DNA template (at ∼1 ng $\mu L^{-1}$) were used in a final volume of 25 $\mu L$ with 0.2 $\mu M$ concentrations of each primer. The PCR cycling protocol consisted of 35 cycles of denaturation (95 °C for 30 s), annealing (54 °C for 30 s) and elongation (72 °C for 1 min) with an initial denaturation step (95 °C for 4 min) and a final elongation step (72 °C for 7 min). Two independent PCRs representing technical replicates were performed for each sample. The RNA was amplified by performing RT-PCRs using the Titan One Tube RT-PCR kit (Roche Molecular Systems). The reaction was carried out in a final volume of 50 $\mu L$ with 0.2 $\mu M$ of each primer. Reverse transcription (42 °C for 30 min) was followed by PCR amplification with an initial denaturation step at 94 °C for 3 min, 38 cycles of 30 s at 94 °C, 30 s at 54 °C and 45 s (+5 s/cycle from the 11th to the 38th cycle) at 68 °C and a final elongation at 68 °C for 7 min. Two independent RT-PCRs were performed for each sample. The quality of the PCR products was checked on High Sensitivity DNA chips (Agilent, Santa Clara, CA, USA) and purified amplicons were quantified by spectrofluorometry using the Quant-iT PicoGreen dsDNA Assay kit (Invitrogen). The libraries were pooled in equimolar amounts and purified using the AMPure XP system (Beckman-Coulter, Brea, CA, USA). Any traces of concatemerized primers were removed by subjecting the libraries to microelectrophoresis on a Caliper Labchip XT instrument (Perkin Elmer, Waltham, MA, USA). The libraries were amplified by emPCR using the GS FLX Titanium MV emPCR Kit with Lib-L chemistry and sequenced on a GS FLX+ sequencer (Roche/454, Branford, CT, USA) following the manufacturer's instructions.

## Sequence analysis using a dedicated automated pipeline

Quality trimming and filtering of amplicons, OTU identification, and taxonomic assignments were carried out with a combination of publicly available sequencing data analysis tools (Cutadapt, Mothur, Dnaclust) and in-house python scripts within a Galaxy instance at the Genouest platform (http://www.genouest.org/), as described elsewhere (*Ciobanu et al., 2014*; *Nunes et al., 2015*; *Ben Maamar et al., 2015*). Briefly, sequences shorter than 200 bp in length, with homopolymers longer than 8 bp or with ambiguous nucleotides, were removed from the dataset. Sequences containing errors in the MID or primer sequences were discarded. Chimeric sequences were eliminated using the chimera.uchime command of the Mothur tool suite. It is well known that both PCR and

pyrosequencing can induce erroneous sequences (*Shakya et al., 2013b*) leading to poor diversity estimates. To improve sequence quality, two independent PCR reactions were performed for each sample and replicates were sequenced. Only sequences displaying 100% identity among these technical replicates were retained for subsequent analysis. Sequences were grouped into OTUs with a sequence identity threshold of 97%. Consequently one OTU was defined by at least two identical sequences originating from technical replicates. The taxonomic affiliations of the sequences and OTUs were determined by comparison with the SILVA database 115 (*Quast et al., 2013*). Two OTUs, representing 1,132 sequences, assigned to the *Chloroplastida,* were removed. The study accession number in the European Nucleotide Archive is PRJEB12655.

## Statistical and diversity analyses

The samples were normalised to 1,288 sequences for analyses of alpha and beta diversity whereas the full dataset was used to analyze the core microbiome in co-occurring plants within a single population. All statistical analyses were performed in R (*R Core Team, 2013*) using the VEGAN package (*Oksanen et al., 2011*). To check the sequencing depth, rarefaction curves were computed using the rarefaction function. Alpha diversity and richness were estimated for each sample using Hill diversity numbers (*Hill, 1973*) and the Chao 1 index (*Chao, 1984*). Hill diversity numbers allow accurate comparison of species/OTU diversity across samples. The significance of the Hill diversity numbers depends on the value of the $q$ parameter in the Hill formula. This parameter allows species to be weighted more or less equally. For $q = 0$, OTUs or species are weighted equally and the Hill diversity is equal to the OTU richness while for $q = 1$, the Hill diversity corresponds to the Shannon diversity and for $q = 2$, the Hill diversity is equal to the Simpson index. One community can be considered more diverse when all of its Hill diversity numbers are higher than those of the other communities. The Chao 1 index estimates the unseen diversity by taking rare OTUs into account. The alpha diversities and taxonomic compositions of the DNA and RNA fractions were compared using paired Student's $t$-tests or Wilcoxon rank-sum tests only when the alpha diversity values did not follow a normal distribution or were heteroscedastic.

Beta diversity was studied by using non-metric multidimensional scaling (NMDS) with the Bray–Curtis dissimilarity matrix. Data were transformed using the square root and the Wisconsin double standardization implemented in the metaMDS function. Procrustes analysis was conducted on the NMDS scores to assess the concordance between the communities in the DNA and RNA fractions. Significance of the concordance was tested by permutation (10,000) using the protest function (*Peres-Neto & Jackson, 2001*).

We defined the core microbiome as the proportion of OTUs shared by the studied co-occurring *A. stolonifera* plants. To determine whether the number of shared OTUs was dependent on the sampling effort, we performed random re-sampling and increased the sampling size from one to the total number of samples for each fraction. We defined the 'DNA core' and the 'RNA core' as the proportion of OTUs present in the DNA and the RNA fraction respectively of all samples.

## Phylogenetic tree construction

Representative sequences of OTUs were aligned using SINA aligner v1.2.11 (*Pruesse, Peplies & Glockner, 2012*), imported from the non-redundant SILVA SSU Ref ARB database (release 115). Alignments of the reference sequences and of representative OTU sequences were exported from ARB (*Ludwig et al., 2004*). Gaps and ambiguously aligned positions were excluded. One alignment was obtained for each main fungal phylum. The model of sequence evolution that best fitted the aligned sequences was selected using jModelTest v2.1.4 (*Darriba et al., 2012*). Phylogenetic trees were constructed by maximum likelihood using TREEFINDER and by bayesian inferences using MrBayes v.3.2.2 (*Ronquist et al., 2012*). Maximum likelihood bootstrap values were calculated from 1,000 replicates and Bayesian posterior probabilities were calculated using 100,000 generations sampled every 100 generations.

## Phylogenetic structure

The community phylogenetic structure was studied by applying the Picante package (*Kembel et al., 2010*) to the global phylogenetic tree including all the fungal OTUs. Comparison of the phylogenetic relatedness of the OTUs in the DNA and RNA fractions of each sample with the local pool of OTUs (OTUs found in all samples) was based on two indices: the mean pairwise distance (MDP) and the mean nearest taxon distance (MNTD) described by *Webb et al. (2002)*. The phylogenetic distances measured in the observed communities were compared with those in the null communities generated by randomization (1,000 permutations) by calculating standardized effect sizes (SES). Tip labels were randomly shuffled across the tips of the phylogeny for the null model. In this model, the community data matrix is not randomized in order to fix most of the patterns (species richness, observed occupancy rates and OTU abundance) and to allow the pattern of interest to vary (phylogenetic distance) (*Swenson, 2014*). However, type I error rates may be inflated if a phylogenetic signal in OTU abundance occurs (*Hardy, 2008*). As no phylogenetic signals in OTU abundance were found using Blomberg's $K$ statistic (*Blomberg, Garland & Ives, 2003*) the use of this null model was validated. Negative SES values and $p < 0.05$ indicated phylogenetic clustering and smaller than expected phylogenetic distances among co-occurring OTUs. Two analyses were performed to create the null communities, one using all the OTUs and another one using the OTUs obtained separately from the DNA and the RNA fractions. Both methods gave similar results and only results from the former analysis are therefore presented.

## Phylogenetic signal

Here we measured the '*mean expression ratio*' of each OTU, in order to access the metabolic status of the microorganisms. This metric was obtained for each OTU by dividing the mean relative abundance in the RNA fractions (*mean $RA_{RNA}$*) by the sum of the mean relative abundance in the DNA (*mean $RA_{DNA}$*) and RNA fractions among all samples.

$$mean\ expression\ ratio = \frac{mean\ RA_{RNA}}{mean\ RA_{RNA} + mean\ RA_{DNA}}$$

with

$$mean\ RA = \frac{\sum_{i=1}^{n} RAOTUi}{n}$$

$n =$ number of plants showing the OTUi within their DNA and/or RNA fraction The standard errors were then calculated for each OTU of the 19 samples. A zero value for this ratio would indicate an absence of the OTU from the RNA fractions while a value of one would indicate that the OTU was not detected in the DNA fractions. To determine whether the '*mean expression ratio*' across the phylogeny was not random, the phylogenetic signal was calculated using the *Kse* statistic (*Blomberg, Garland & Ives, 2003*; *Ives, Midford & Garland, 2007*) for the global phylogenetic tree and for phylogenies built for each main fungal phylum (i.e., Glomeromycota, Basidiomycota, and Ascomycota). This index integrates measurement errors of the trait by calculating the standard errors of the means for each OTU. The *Kse* statistic compares the observed phylogenetic signal in the '*mean expression* ratio' to the signal obtained under a Brownian motion model of trait evolution. In this model, traits evolve as a random walk along the branch length of a phylogeny. The statistical significance of *Kse* was evaluated by comparing the observed patterns of variance of the independent contrasts of the trait to a null model. Taxa labels were shuffled across the tips of the phylogeny for the null model. A '*mean expression ratio*' with $p < 0.05$ indicated a non-random phylogenetic signal.

## RESULTS

### DNA- and RNA-based analyses provided similar alpha and beta diversity profiles but different OTU richness levels

We examined alpha and beta diversity measures of the fungal community colonizing the roots of 19 *Agrostis stolonifera* plants to see whether the nature of the co-extracted nucleic acids encapsulated different information. The mean Chao 1 index, that estimates 'true OTU diversity' by taking rare OTU into account, did not differ significantly ($p = 0.11$) between the DNA and RNA fractions (S.Chao1$_{DNA}$ = 215, S.Chao1$_{RNA}$ = 232). On average, the communities in the RNA fractions were not considered more diverse than those in the DNA fractions as not all of the Hill diversity numbers were significantly different (Fig. 1A). Nevertheless, sample-to-sample variations did exist (Fig. 1A). Interestingly, OTU richness, i.e., a special case of Hill diversity numbers when $q = 0$, was significantly higher in the RNA fractions than in the DNA fractions, with on average 149 OTUs in the RNA fractions and 133 OTUs in the DNA fractions ($p < 0.01$) (Fig. 1A). The sequencing depth was similar between both fractions (Fig. S1).The procrustes analyses indicated a significant similarity ($p < 0.001$) between the distances obtained from the DNA and RNA matrices. Thus, the beta diversity was not structured according to the type of nucleic acid.

### Nucleic acid type impacts the perception of the taxonomic composition

The fungal communities in the roots were dominated by *Pezizomycotina* (Ascomycota) and *Agaricomycotina* (Basidiomycota) (Fig. S2). Each phylum, except for the Basidiomycota, was dominated by the same OTU in both the DNA and the RNA fractions, representing

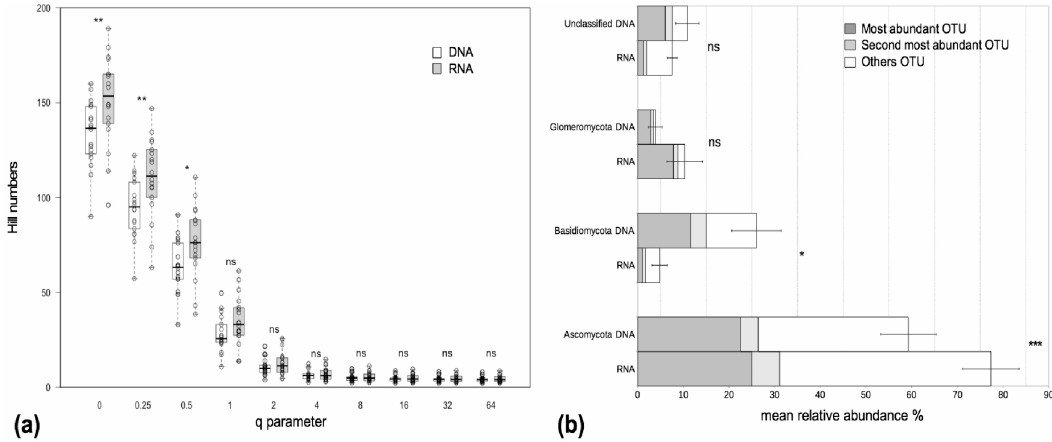

(a)                                              (b)

**Figure 1** **Alpha diversity and taxonomic composition of fungal communities in DNA and RNA fractions.** (A) Distribution of Hill diversity numbers in DNA and RNA sequence data analyses. Different commonly used diversity indexes are special cases of Hill numbers (e.g., $q = 0$ corresponds to the species richness S, $q = 1$ corresponds to the exponential of the Shannon-Wiener diversity index, $q = 2$ corresponds to the inverse Simpson index). Asterisks indicate significantly different means between DNA and RNA fractions (paired $t$-tests). 'ns': $p > 0.05$, '*': $p < 0.05$, '**': $p < 0.01$. (B) Taxonomic composition of fungal communities in DNA and RNA fractions. Assignation of sequence taxonomy using SILVA database 115. Average relative sequence abundance (% ± se) of each phylum in the fungal kingdom for 19 samples. Asterisks indicate significantly different means between DNA and RNA fractions. 'ns': $p > 0.05$, '*': $p < 0.05$, '***': $p < 0.001$. The mean relative sequence abundances of the two most abundant OTUs are shown for each phylum. The most abundant OTUs in the DNA and RNA fractions were the same except for Basidiomycota. For this latter group, the second most abundant OTU in the RNA fraction was the most abundant OTU in the DNA fraction. There was just a slight difference in sequence number between the first and second most abundant OTU in the RNA fraction (420 and 322 sequences, respectively) and these two OTUs were assigned to the *Agaricomycetes* class.

17%–76% of the total sequence number (Fig. 1B). The most abundant OTUs in the DNA and RNA fractions were the same except for the Basidiomycota. In this phylum, the two most abundant OTUs in the DNA and RNA fractions were assigned to the Agaricomycetes. The most abundant OTUs were assigned unambiguously to the class level (Sordariomycetes) within the Ascomycota, to the order level (Agaricales) within the Basidiomycota and to the family level (Glomeraceae) within the Glomeromycota (i.e., unknown at lower taxonomic rank). The mean relative sequence abundance assigned to the Basidiomycota was significantly higher in the DNA fraction than in the RNA fraction ($p < 0.05$, Fig. 1B). Conversely, sequences assigned to the Ascomycota were more abundant in the RNA fraction than in the DNA fraction ($p < 0.001$, Fig. 1B). Glomeromycota represented 3.9% and 10.1% of the total number of sequences in the DNA and RNA fractions respectively but the differences between these fractions were not significant (Fig. 1B).

## At the population level, very few fungal OTUs were shared by all co-occurring plants

Nine OTUs out of a total of 635 OTUs were found in both the DNA and RNA fractions of all of the 19 co-occurring *A. stolonifera* plants, accounting for 33% of the total number of sequences (Fig. 2). The number of sequences was equally divided into the RNA fraction

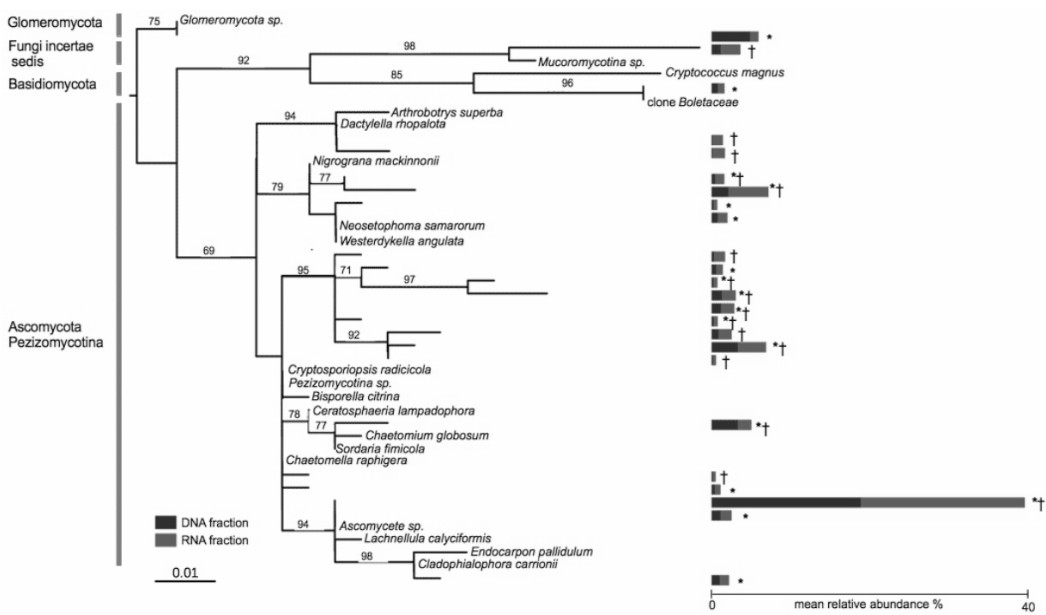

**Figure 2** **Phylogenetic tree of the core microbiome OTUs at the level of 19 co-occurring *Agrostis stolonifera* plants.** Tree construction was based on maximum likelihood. Only bootstrap values above 50 are indicated. The tree was constructed using sequences representative of the OTUs (taxa without names) and the closest reference sequences (taxa names in italic) from the non-redundant SILVA SSURef ARB database (release 115). '*': taxa belonging to the 'DNA core', ' †': taxa belonging to the 'RNA core'. Stacked bars indicate the mean relative abundance of each taxon in the DNA (blue) and RNA (red) fractions of the 19 samples. Some taxa belonging to the 'DNA core' are also found in the RNA fractions but not in all samples and reciprocally.

(54%) and the DNA fraction (46%). Considering each fraction separately, 17 and 16 OTUs in the DNA and RNA fractions respectively were shared by all samples. All OTUs in the 'DNA core' were found in the RNA fraction of at least one sample and reciprocally. The number of OTUs shared by all samples in both fractions was negatively linked to the sampling size (Fig. S3). However, this number of shared OTUs stabilized at around 15 samples, indicating that the 'core microbiome' size would not be diminished by increased sampling within these co-occurring *A. stolonifera* (Fig. S3). The phylogenetic analysis (Fig. 2) revealed that the fungal 'core microbiome' was dominated by Ascomycota and the closest known sequence of the most abundant OTU was designated *Ascomycete sp.* (i.e., an unknown OTU) (Fig. 2). Strikingly, most of the OTUs in the fungal potential 'core microbiome' were only distantly related to known species (Fig. 2).

## The fungal root microbiome is phylogenetically structured

We tested our working hypothesis that the host plant acts as an environmental filter on microbial communities by examining the phylogenetic structure of the fungal root microbiome. For this we measured the standardized effects size of the mean pairwise distance ($SES_{MPD}$) and of the mean nearest taxon distance ($SES_{MNTD}$) in the DNA and RNA fractions of 19 samples (i.e., 38 fractions). Negative values of $SES_{MPD}$ indicated that the co-occurring fungal OTUs were more closely related than expected under a null model,

**Table 1** Standardized effect sizes of the mean pairwise distance values (SES$_{MPD}$) and standardized effect sizes of the mean nearest taxon distance values (SES$_{MNTD}$) for the 19 fungal communities in *Agrostis stolonifera* roots detected in the DNA and RNA fractions.

| | nb OTU | SES $_{MPD}$ | | SES $_{MNTD}$ | |
| Sample | DNA/RNA | DNA | RNA | DNA | RNA |
| --- | --- | --- | --- | --- | --- |
| 1 | 137/121 | −0.14 | 1.27 | −3.45$^*$ | −3.48$^*$ |
| 2 | 170/185 | −2.85$^*$ | −0.18 | −3.92$^*$ | −1.96$^*$ |
| 3 | 98/137 | −4.78$^*$ | −6.31$^*$ | −2.35$^*$ | −4.1$^*$ |
| 4 | 113/176 | −1.09 | −1.65 | −3.27$^*$ | −2.15$^*$ |
| 5 | 132/180 | −0.58 | −2.13$^*$ | −3.23$^*$ | 0.02 |
| 6 | 144/180 | −1.09 | −2.77$^*$ | −1.87$^*$ | −2.77$^*$ |
| 7 | 93/145 | −3.42$^*$ | −3.53$^*$ | −2.82$^*$ | −1.53 |
| 8 | 111/100 | −6.02$^*$ | −3.97$^*$ | −4.2$^*$ | −2.2$^*$ |
| 9 | 87/160 | −1.51 | −1.76$^*$ | −3.71$^*$ | −3.85$^*$ |
| 10 | 124/102 | −2.45$^*$ | 0.78 | −2.61$^*$ | −2.95$^*$ |
| 11 | 157/175 | 0.31 | −0.18 | −1.72$^*$ | −2.02$^*$ |
| 12 | 93/197 | −0.45 | −1.05 | −3.9$^*$ | −2.3$^*$ |
| 13 | 87/165 | −1.7$^*$ | −1.81$^*$ | −3.71$^*$ | −2.26$^*$ |
| 14 | 123/155 | −1.36 | 0.04 | −3.45$^*$ | −2.74$^*$ |
| 15 | 154/164 | −1.51 | −1.55 | −3.96$^*$ | −3.23$^*$ |
| 16 | 138/126 | −2.14$^*$ | −1.83$^*$ | −4.46$^*$ | −4.13$^*$ |
| 17 | 88/142 | −1.35 | −2.13$^*$ | −2.82$^*$ | −2.46$^*$ |
| 18 | 152/104 | −2.64$^*$ | −2.38$^*$ | −3.81$^*$ | −2.16$^*$ |
| 19 | 157/113 | −2.59$^*$ | −2.83$^*$ | −4.33$^*$ | −3.26$^*$ |

**Notes.**

nb OTU: number of OTU in the community excluded reference species.

$^*$Community significantly structured ($P < 0.05$).

and this phylogenetic pattern was significant ($p < 0.05$) in 20 out of the 38 DNA and RNA fractions (Table 1). This phylogenetic clustering based on the SES$_{MPD}$ measures was reinforced by the SES$_{MNTD}$ values (Table 1). The obtained pattern was significant for 36 DNA and RNA fractions indicating that phylogenetic clustering mainly concerned the leaves of the phylogeny (Table 1). The SES$_{MNTD}$ index calculates whether the closest related OTUs (nearest taxon) tend to co-occur or not in the communities relatively to the null distribution. Significant SES$_{MNTD}$ values and non-significant SES$_{MPD}$ values indicated that the OTUs were phylogenetically structured near the tips of the phylogeny and randomly distributed across the tree, i.e., deeper branches contributed less to the pattern.

## Phylogenetic signal

The fungal root microbiome displayed phylogenetic clustering within each sample compared to the local pool of OTUs. We investigated whether this phylogenetic relatedness was linked to ecological similarity by searching for a phylogenetic signal, using the '*mean expression ratio*' as a proxy for a functional trait. Sample-to-sample variations were taken into account in the *Kse* index. We found that our trait, i.e., the '*mean expression ratio*', displayed a significant phylogenetic signal (Table 2). This was true for the global phylogeny including all the fungal OTUs and also for the phylogenies built for each fungal phylum

**Table 2  Phylogenetic signal in phylogenies.**

| Phylum | *Kse* |
|---|---|
| Ascomycota | 0.41[**] |
| Basidiomycota | 1.08[***] |
| Glomeromycota | 0.72[***] |
| All Fungi | 0.39[**] |

Notes.
[**] $p < 0.01$.
[***] $p < 0.001$.

(i.e., Glomeromycota (Fig. S4), Basidiomycota (Fig. 3), and Ascomycota (Fig. S5). Thus, the '*mean expression ratio*' was linked to phylogenetic relatedness. The *Kse* statistic was less than one for all phylogenies except the Basidiomycota phylogeny (Table 2). *Kse* values below one indicated that the OTUs were more divergent in their '*mean expression ratio*' than would be expected under a Brownian motion model of trait evolution whereas a value above one indicated that the values of the 'mean expression ratio' were more similar between phylogenetically related OTUs than between OTUs drawn at random.

## DISCUSSION

In this study we assessed the complexity of the fungal root microbiome of co-occurring *A. stolonifera* and tested the hypothesis that fungal communities in plant roots are not random assemblages: we demonstrated for the first time that the phylogenetic signal of the fungal root microbiome was phylum dependent. This new understanding resulted from the application of DNA and RNA co-extraction strategy.

### OTU richness and taxonomic composition of the root fungal microbiome differed between the DNA and RNA fractions

In previous studies, the fungal community composition in roots was essentially analyzed either by cloning of PCR products (e.g., *Vandenkoornhuyse et al., 2002a*; *Vandenkoornhuyse et al., 2002b*) or more recently, by mass sequencing of amplicons using targeted DNA (e.g., *Opik et al., 2013*; *Shakya et al., 2013a*; *Bonito et al., 2014*). Others studies based on RNA and DNA extractions have used the RNA/DNA ratio as a proxy to investigate microbial dormancy and to estimate the metabolically active community (*Aanderud et al., 2016*; *Jones & Lennon, 2010*). In a similar way we defined the root fungal microbiome of co-occurring *A. stolonifera* plants by combining the DNA- and RNA-based 18S rRNA amplicon analyses. However, the nucleic acids were co-extracted in order to limit any extraction-related bias. We found higher OTU richness levels in the RNA fractions than in the DNA fractions (Fig. 1A, $q = 0$), that was consistent with the study by *Kuramae et al. (2013)* and indicated that RNA-based approaches capture higher numbers of different OTUs in a given sample. At first glance this may seem surprising, but considering that RNA extracts contain roughly 80–90% of rRNA transcripts, the rRNA genes were much more diluted in the DNA fraction and could not be amplified. Thus, for a given quantity of nucleic acids, a higher abundance of targeted 18S rRNA genes was introduced into the RT-PCR mix possibly allowing the capture of more diverse molecules. Although not significantly different, the Chao 1 index

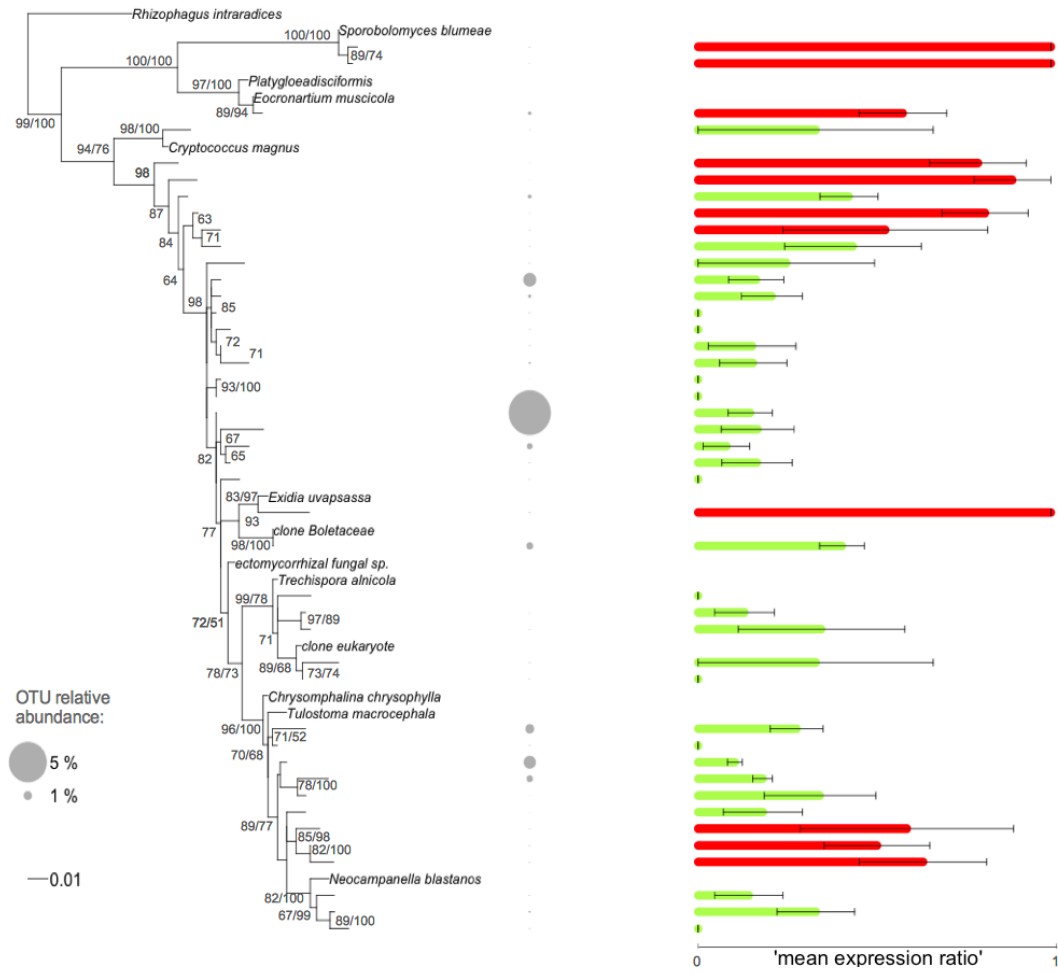

**Figure 3  Phylogenetic tree of the Basidiomycota related root fungal microbiome OTUs.** ML tree based on 432 bp of SSU rRNA gene sequences amplified from roots of *Agrostis stolonifera*. The tree was constructed using representative sequences of the OTUs (taxa without names) and the closest reference sequences (taxa names in italic) from the non-redundant SILVA SSURef ARB database (release 115). Barplots represent the mean expression ratio for each OTU among all samples. Null values indicate that this OTU was not detected in the RNA fraction, value = 1 indicates that this OTU was not detected in the DNA fraction, value = 0.5 indicates that the sum of the relative abundance between DNA and RNA fractions was equal. Green bars: values below 0.5, red bars: values ≥ 0.5. Error bars indicate ±SE. Grey circles indicate the relative abundance of each OTU in the whole dataset. Node support values above 50 are given in the following order: bootstrap values and Bayesian posterior probabilities.

exhibited the same trend of decreased richness in the DNA fraction. When other alpha diversity indexes were used, the differences were also not significant (Fig. 1A, $q > 0.5$). However, significant differences between the DNA and RNA-based analyses were detected when taxonomic composition was taken into account (Fig. 1B). The relative sequence abundance of the Basidiomycota was higher in the DNA fractions than in the RNA fractions and the opposite was true for the Ascomycota (Fig. 1B). It is tempting to conclude that the Basidiomycota was less 'active' than the Ascomycota phylum. However, while RNA represents expressed gene levels and is the best proxy for actively growing microorganisms

at the nucleic acid level, there is not always a direct link between rRNA abundance, growth and activity in microorganisms (*Blazewicz et al., 2013*). The absence of a given OTU from the RNA-based analysis, but detected in the DNA, was not related to amplification bias (i.e., only OTUs found in the true replicates were kept). One possible explanation was that the OTUs detected from DNA fraction originated in part from dormant or dead cells.

These results show that it is essential to be aware of the type of nucleic acid being used for comparisons or meta-analyses and highlights the advantages and importance of taking both kinds of nucleic acids into account to gain deeper insights into microbial diversity.

## Fungal root microbiome is diverse and highly variable among co-occurring *A. stolonifera*

In the nineteen *A. stolonifera* plants, harvested from a natural ecosystem, we found 635 OTUs with a dominance of *Pezizomycotina* (Ascomycota) followed by *Agaricomycotina* (Basidiomycota). The Glomeromycota represented only a small proportion of the total number of OTUs (less than 4%) in the DNA fraction, which is comparable to the fraction found in Agave species (*Coleman-Derr et al., 2015*). However, this proportion was higher in RNA fractions (10%), underlining the interest of RNA based-analyses. A few published studies have focused on the fungal fraction of the root microbiome, mainly in woody plant species, by mass sequencing from DNA extracts (*Gottel et al., 2011*; *Shakya et al., 2013a*; *Bonito et al., 2014*; *Coleman-Derr et al., 2015*). Although these studies involved different primer pairs and different hosts, the fungal taxonomic composition at the subphylum level was similar to our study. The fungal 'core microbiome' at the *A. stolonifera* population level (i.e., core microbiome *sensu Vandenkoornhuyse et al., 2015*) accounted for ∼1.5% of the total OTUs (i.e., 9/635). The core microbiome is expected to decrease in complexity whereas the level of ecological organization will increase (i.e., from individual to ecosystem) (*Vandenkoornhuyse et al., 2015*). Although the level used to define the 'core microbiome' was low (i.e., a population of co-occurring plants) and no temporal changes were taken into account, the complexity of the fungal 'core microbiome' was lower than expected given the fact that all the *A. stolonifera* sampled in this study came from the same turf. Increased sampling within this *A. stolonifera* population would be unlikely to modify the number of shared OTUs (Fig. S3), but might increase the total number of OTUs detected. Moreover, the OTUs shared by all the plants in both their DNA and RNA fractions were mainly abundant OTUs affiliated to *Pezizomycotina* (Ascomycota), which represented more than 30% of all the sequences.

An increased sequencing depth would lead to an increase of the rare, rather than abundant, OTUs. Unexpectedly, only 1 OTU affiliated to the Glomeromycota forming arbuscular mycorrhiza with *A. stolonifera* was shared by all individuals, while most of the other OTUs shared by all individuals were unknown Ascomycota forming unknown fungal endophytes (Fig. 2). As the different Ascomycota and Basidiomycota fungal endophytes are acknowledged to be involved in plant resistance to stresses (e.g., *Soares et al., 2016*; *Cosme et al., 2016*), our findings raise important new questions about the functions of these unknown endophytes and more widely emphasize the need for a more holistic perception and understanding of the plant holobiont (*Vandenkoornhuyse et al., 2015*).

### *A. stolonifera* fungal endophytic communities exhibited significant phylogenetic clustering

We found that the fungal communities in plant roots exhibited a significant phylogenetic structure whether the analyses were based on DNA or RNA which confirmed our working hypothesis that the fungal communities in plant roots are not random assemblages (Table 1). At the host-plant scale, the phylogenetic structure of the fungal microbiome showed greater clustering than would be expected under a null model. Thus, the fungal OTUs in an individual plant were more closely related to each other than to the pool detected at the community level. It is not easy to discriminate between the different neutral or ecological processes underlying this pattern because various explanations are feasible (*Losos, 2008*; *Revell, Harmon & Collar, 2008*). Indeed, Phylogenetic clustering can be generated by dispersal limitation as well as by inter-specific interactions or environmental filtering (*Bell, 2005*; *Cavender-Bares et al., 2009*; *Helmus et al., 2007*). However, the spatial scale of our study was small in relation to the known dispersal capacities of fungi (*Taylor et al., 2006*). Dispersal limitation was therefore unlikely or, at least, would not be the main driver of the observed phylogenetic pattern. Ectomycorrhizal fungal communities in plant roots have been shown to be shaped by competitive interactions (*Pickles et al., 2012*) and host-plant specificity or preference has also been reported (*Ishida, Nara & Hogetsu, 2007*; *Tedersoo et al., 2013*). Similarly, assembly of the arbuscular mycorrhizal fungal community in plant roots reflects a degree of host-plant preferences (*Vandenkoornhuyse et al., 2002b*; *Vandenkoornhuyse et al., 2003*; *Davison et al., 2012*). In *Populus deltoides* roots, the endophytic fungal communities were found to be clearly distinct from the surrounding rhizospheric communities (*Gottel et al., 2011*; *Shakya et al., 2013a*). Environmental filtering by the host, which would represent a specific habitat, could thus be an important mechanism leading to phylogenetic clustering. Assuming that phylogenetically close OTUs share common phenotypic traits (*Webb et al., 2002*), the host plant would select for OTUs with particular biological features, adapted to a symbiotic life style. This implies that traits favouring symbiosis would be phylogenetically conserved in fungi. Environmental filtering is probably not an exclusive mechanism and other processes, such as inter-specific microbial interactions, might explain the observed phylogenetic structure. For instance, if competitive ability is assumed to be correlated with phylogenetic distance then competition can drive phylogenetic clustering (*Mayfield & Levine, 2010*). Mechanisms leading to an over-dispersed phylogenetic structure, especially inter-species competition, which classically leads to species exclusion between relatives (*Diamond, 1975*), would have a weaker effect than environmental filtering by the host. Kin selection, a strategy favouring the reproductive success of an organism's relatives, might also be important in structuring fungal communities.

### Disentangling the processes underlying the observed phylogenetic structure

The relative importance of the above-mentioned processes in explaining the observed phylogenetic clustering can be elucidated by detecting the phylogenetic signal of relevant trait(s) (*Mayfield & Levine, 2010*). However, phenotypic traits are especially difficult to

measure in complex communities, particularly for uncultivated microbes. In this study, we used the 'mean expression ratio' to access the microbial metabolic status. Sample-to-sample variations were taken into account using the *Kse* index that allowed the power of the test to be increased and avoided an estimation bias (*Hardy & Pavoine, 2012*). Interestingly, a significant phylogenetic signal was found for all fungal groups (Table 2). Within the Ascomycota and Glomeromycota, the 'mean expression ratios' of evolutionary-related OTUs were more divergent than would be expected under a Brownian motion model of trait evolution (Table 2, Figs. S4, S5). Conversely, the OTUs in the Basidiomycota were more similar than expected, i.e., related OTUs shared similar expression ratios (Table 2 and Fig. 3). Thus, the combined information about the phylogenetic structure (Table 1) and the phylogenetic signals analyses (Table 2) suggested that the Basidiomycota assemblage was mainly governed by environmental filtering, favouring the co-existence of related and similar OTUs in their 'mean expression ratio'. In contrast, the Ascomycota and Glomeromycota assemblages were more impacted by competitive interactions promoting the co-existence of phylogenetically related but dissimilar OTUs in their 'mean expression ratio'. In this latter case, competitive ability would be positively correlated with phylogenetic distance or, competition would drive character displacement rather than OTUs exclusion. Other relevant functional traits now need to be used to confirm our interpretations and to improve our detection and understanding of the phylogenetic signal. One possibility would be to perform a comparative transcriptomic analysis of the fungal root microbiome, although a number of methodological padlocks would first need to be broken, notably the selective extraction of fungal RNA.

## Conclusion and prospects

We describe here the root fungal microbiome associated with an *A. stolonifera* population using a molecular strategy combining DNA- and RNA-based approaches. We were able for the first time to draw up a comprehensive picture of the phylogenetic patterns existing in the fungal root microbiome by examining the phylogenetic structure and measuring the phylogenetic signal. We found that the fungal communities were not randomly assembled but instead appeared to be specifically filtered by their plant host. We thus provide new insights into the rules of assembly governing the root fungal microbiome community. The limited number of OTUs shared by all individuals and the clustered phylogenetic structure suggested that each plant recruits a particular microbial community to adapt to environmental conditions at a microscale.

The use of evolutionary information to describe an ecological pattern is a first step towards a full understanding of community assembly. Further experimental studies are now needed to focus on the processes underlying these patterns.

## ACKNOWLEDGEMENTS

We are very grateful to Diana Warwick and the four anonymous reviewers for comments and suggested modifications on a previous version of the manuscript, to the Biogenouest Genomics and the Human and Environmental Genomics platform (https://geh.univ-rennes1.fr/) for the sequencing, and to the Genouest Bioinformatics facilities.

### Funding

This work was supported by a grant from 'l'Agence Nationale de la Recherche' (ANR-10-STRA-0002). The funders had no role in study design, data collection and analysis, decision to publish, or preparation of the manuscript.

### Grant Disclosures

The following grant information was disclosed by the authors:
'l'Agence Nationale de la Recherche': ANR-10-STRA-0002.

### Competing Interests

The authors declare there are no competing interests.

### Author Contributions

- Amandine Lê Van conceived and designed the experiments, performed the experiments, analyzed the data, wrote the paper, prepared figures and/or tables, reviewed drafts of the paper.
- Achim Quaiser and Philippe Vandenkoornhuyse conceived and designed the experiments, analyzed the data, wrote the paper, reviewed drafts of the paper.
- Marie Duhamel conceived and designed the experiments, contributed reagents/materials/analysis tools, reviewed drafts of the paper.
- Sophie Michon-Coudouel performed the experiments, contributed reagents/materials/analysis tools, reviewed drafts of the paper.
- Alexis Dufresne conceived and designed the experiments, analyzed the data, contributed reagents/materials/analysis tools, wrote the paper, reviewed drafts of the paper.

### DNA Deposition

The following information was supplied regarding the deposition of DNA sequences:
European Nucleotide Archive PRJEB12655.

### Supplemental Information

Supplemental information for this article can be found online at http://dx.doi.org/10.7717/peerj.3454#supplemental-information.

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
