# Peer review of "Ecophylogeny of the endospheric root fungal microbiome of co-occurring Agrostis stolonifera"

_PeerJ, doi:10.7717/peerj.3454_

## Round 0.1 · original submission · Major Revisions

As you can, see the paper has been extensively reviewed, and each of the reviewers has made substantive comments that will require detailed and careful consideration.

The extent and nature of the revisions requested collectively have left me in some doubt as to whether or not I should reject the paper at this point. However, I am bound to say that none of the reviewers recommended an outright rejection, and they clearly find some merit in the study. Nevertheless, it is a close call, and I would suggest that you and the co-authors should fully reappraise the paper in the light of the reviewers' criticisms, particularly with regard to the presentation and interpretation of results, as well as the justification for your conclusions.

Two major over-arching issues, amongst others highlight in the review process, that I would ask you to address directly, if you decide to proceed with a revision are: (1) to provide a clear and sufficiently detailed description of your experimental design and experimental approach; and (2) to ensure that there is a much more critical and better justified interpretation of the results, which takes into account the limitations and biases of the experimental approach (e.g. problems with depth of sequencing and OTU definition; and the limited sampling of Agrostis stolonifera roots from just one, modified or 'acclimated' sample of turf). In these two respects, I fully concur with the criticisms of Reviewers 3 (see also annotations on MS) and 4.

Finally, I would also ask you to reconsider the presentation of all of the paper's figures, to ensure that the reader has all of the information to enable them to follow your interpretations and to provide clear justifications for your main conclusions. In the context of a revised and focused paper, please would you also carefully consider whether or not all of the figures and tables are necessary.

Reviewer 1 ·

Basic reporting

The English is largely good; some corrections indicated in Section 4. The introduction, structure and figures are all of a good standard. However, I could not see any raw data supplied.

Experimental design

The study addresses an interesting research question, comparing host-mediated versus competitive influences on mycorrhizal communities. It also compares the use of DNA and RNA amplicon data for community characterisation. The study appears to have been well-conducted, and the methods well reported. I have two main queries concerning the methods: firstly, whether the analysis pipeline is available for other researchers to use. The second concerns the metric ‘mean expression ratio’ that the authors have developed. It is central to the paper but, from the description provided, I wasn’t clear how it was calculated.
I do not have enough experience of phylogenetic analysis to comment on the details of the methods for this aspect, e.g. use of a Brownian motion model of trait evolution.

Validity of the findings

The analysis and conclusions drawn from the data appear to be sound.

Additional comments

Specific comments are given below with line numbers:
33-34: “show a phylogenetic structure that can be explained…”
36: “phylum” (phyla is the plural)
49: In this context, it is preferable not to cite an unpublished paper.
50: “controls”
52: Why is the endosphere a restricted area?
78: “Conversely” would be more standard English here.
79-81: How would one ascertain whether that assumption had been satisfied?
87: Here and throughout, ‘host plant’ should not be hyphenated when ‘plant’ is used as a noun; it should only be hyphenated when both words are being used together as an adjective e.g. ‘host-plant specificity’.
91: How is this a proxy for a trait? Is it not a proxy for activity?
107-111: Where were the turfs acclimated and how?
115: “into”
118:”RNA and DNA”
124: Please include primer sequences.
155-157: Is this pipeline/scripts available?
162-163: I’m not sure what you mean by true replicates in this context.
178: “accurate comparison of”
177-180: A bit more explanation of the Hill diversity numbers would be beneficial – particularly as later the q-values are discussed without an explanation of what they are.
191: “proportion of OTUs” would be clearer here.
223: More explanation of the mean expression ratio is required here, in particular an equation to show exactly how it is calculated. I could not work out how this was done from the verbal description alone.
262-263: What were these OTUs?
266-268: Please rephrase this sentence to make it clearer.
273: It is preferable to have headings as statements, rather than questions.
277: Were 9 OTUs represented in both the DNA and RNA of all samples, or were, for instance, 4 found in the DNA of all samples and 5 in the RNA of all samples?
297: Can you find another word for fractions here (and line 299)? I couldn’t initially work out what it meant.
299: Why does this indicate that it mainly concerns the leaves, and what exactly do you mean by this phrase? Do you mean that deeper branches contributed less to the pattern?
336: “indicates”
337: “glance this may seem surprising”
350: This doesn’t exclude the possibility of primer bias, though.
372-374: It’s not clear how the results of Blaalid study relate to this discussion.
378: What proportion of RNA transcripts did it account for? Are the core OTUs also the most active?
384: On what was this expectation based?
430-435: Somatic incompatibility is a rather different issue, as it relates to self/non-self recognition – any non-self individual will trigger an antagonistic response.
651: Either “fungal kingdom” or “kingdom Fungi”
Fig 1b: It would be good to see the identity of the ‘most abundant’ and ‘second most abundant’ OTUs.
Fig 2: It is not clear how much of the Ascomycota bar represents the Pezizomycotina.
Fig 3: Why is the tree for the Ascomycota in the Supplementary Material?

Reviewer 2 ·

Basic reporting

The writing of this manuscript generally meets expected professional standards regarding English expression but some language, particularly in the discussion becomes conversational and should be revised for formality and to the standard practice of 3rd person, past tense.

suggestion:
Check for formality of the language (statements such as ‘indeed’ or ‘if this is true’ are more conversational than formal), lines 64,52,85 257, 337

The work has a clearly defined research question “what governs fungal-endophyte-community assembly?”, however the manuscript would benefit from a re-working of the discussion to keep the focus on the scientific question and effectively communicate the ‘take home’ messages.

Suggestions:
First paragraph of discussion should be used to summarise the main results/interpretations rather than recapitulate methods.
Section headings in results and discussion should be carefully considered: ideally I would recommend results headings to be stated as complete premises that encapsulate the main finding.
Lines 327-355: observed differences in DNA and RNA extraction methods, although noteworthy, are not central the core research question as stated in the summary and should be reworked to concisely state the findings and highlight their relevance with respect to the research question.
Line 357: this heading does not reflect the discussion that follows
Line 357 -377: discussion needs to be concisely and clearly linked to the research questions that underlie this work. Connection would be clearer if the results specific to this study were highlighted first and then placed in the context of the wider literature rather than the opposite. The discussion surrounding the general focus of rhizosphere research on bacterial communities does not seem relevant to the discussion of the research question.

Some figures are difficult to interpret and don’t seem to add anything to the manuscript. In general the presentation of the data needs significant improvement:

Figure S1: All figures, including supplemental, should have error bars
Figure 2: The phylogenetic tree contains more tips than are described by the bar charts (?) this makes it is unclear which tree tips relate to the abundance bar charts of the core microbiomes and the there is no explanation regarding the additional tips
Figure 3: I am not convinced that this figure adds to the manuscript. Seems all the relevant information is contained in table 2.
Figure S3: as with Fig 3
Figure 1b: Could you add error bars at the OTU level?
Figure 1 and 3 and tables 1 and 2 are not explicitly referred to in the discussion; the discussion should be amended with appropriate references or the figures discarded if they do not add to the discussion


Other comments:
Lines 48-50: I find this sentence very difficult to understand, pls restated more clearly
Line 56-57: this line could be removed or improved by listing specific examples of the ‘associated microorganisms’
Line 61: insert paragraph break?
Line 61-63: this statement needs to be substantiated with references
Line 84: short sentences here decrease readability
Line 88: a statement such as ‘in this study’ is needed to improve clarity here

Line 109: include a brief description of A.stolonifera beyond its latin name. i.e. grass/herb clumping/rhizomatous growth habit ect
Line 134 and 139: check manuscript for consistency of units (µl versus µL)
Line 148: AMPure XP system
Line 164: change ‘in’ to ‘into’
Line 178: revise sentence re: “…allow to accurately comparing…”
Line 174 – 184: include appropriate references for Hill numbers and Chao indexes
Line 185: add NMDS acronym in parentheses

Line 248: change statistical ‘P’ to ‘p’ throughout manuscript
Lines 263-266: These two sentences are unclear, please revise
Line 273: restate as a premise that encapsulates the result
Line 335: change ‘that is’ to past tense

Experimental design

This research is within the aims and scope of PeerJ. The research question regarding the mechanisms governing microbial community assembly is highly relevant to the field of microbial ecology. I found the experimental methods and design to be robust but have the following questions and comments:
Line 107-110: it would be valuable to know rough dimensions of the collected turf to get a sense of the spatial scale for this experiment. Do the 19 plants (that covered 94% of the turf) constitute all the A.stolonifera that was present in the turf sample or are they a subset; are they spatially disconnected or interconnected?

Line 133: “Two microliters of DNA template were used…” What was the concentration of the template DNA?; were the concentrations of DNA or RNA samples standardized prior to PCR amplification? If not, did you account for amplification differences?

Line 223-227: It is not made clear why the authors have chosen to use the ‘mean expression ratio’ as a proxy for a functional trait. The most in-depth explanation ‘to access the microbial metabolic status’, which appears in line 442 of the discussion, is similarly vague. Please elaborate on why this measure was chosen and how it is proposed to be a proxy for microbial metabolic status.

Validity of the findings

The data from this research was examined in a statistically sound fashion. The conclusions drawn during the discussion are appropriate and relevant but the connection with the core research questions is sometimes lost (See Basic reporting and for suggestions)

Additional comments

I found this research interesting and relevant to the question of community assembly; very neat work. But I struggle to take away anything meaningful from the images of phylogenetic trees. clearly phylogenetic structures are central to the story of this manuscript but is there are better way to present this data or to walk the reader through the patterns/trends they should be observing?

Reviewer 3 ·

Basic reporting

1) Generally, the wording needs to be clarified, especially in the introduction. When I read it, I get the impression that the authors' first language is not English. I sympathize with the difficulty in writing in another language, however some work needs to be done to refine the language to make it more readable.

2) It is very unclear where paragraphs start and finish due to the lack of indentation. This makes it difficult to critically evaluate the writing.

3) Topic sentences need to be used more frequently, in order to make it clear to a reader what the main point of a paragraph is.

4) I don't believe they sufficiently reference existing literature comparing DNA to RNA community sequencing. There are papers in the fungal literature that use DNA and RNA, and there is even more in the bacteria literature. Some main points from this work needs to be acknowledge in the Introduction and more comparisons made in the Discussion. For example, Jay Lennon's lab at Indiana University has done a lot with DNA:RNA community sequencing to understand dormancy and activity in bacteria comunities. Not recognizing this work makes it seem like this paper is doing this for the first time, and it misses out on the opportunity to utilize findings from previous work to build the case for using DNA and RNA together.


5) In the Discussion there is reference to endophyte literature on lines 392/393. But, the articles cited are about systemic endophytes in aboveground tissues in grasses, not about root endophytes. However, the authors discuss this in the context of their data, which is about roots. Please utilize references that more applicable to the topic at hand.

6) I have lots of notes included on the pdf of the manuscript that I submitted with my review. Please take a look at these more minor points.

Experimental design

1) Please see my point about including more discussion on the existing RNA:DNA literature with fungi and bacteria under "basic reporting"

2) Choice of sequence region: I under stand why the 18S was used, it is nicely align-able so phylogenies can be made. However, it does not resolve data down to the level of species. The authors need to clearly define what level of taxonomic resolution their marker region provides information about. This is not provided at any point in the manuscript, but will have a big impact on the inference made from the data. Phylogenetic signal may change when the marker region delineates among species vs some coarser resolution. My belief is that the region used delineates at about Order or Family, maybe. So, the OTUs are composed of many species that could have diverse functions within a family or order.

3) Multiple PCR replicates were performed and sequences were removed if they were not in all replicates. I really like this approach. However, becareful with what you state this does. I think it mostly reduced stochastic artifacts that might only occur in one sample. But, it does not get rid of artifacts that could happen easily, such as some chimeras, and it does not eliminate taxonomic bias that might be due to primers favoring some taxa over others do to slight mismatches or due to amplicon length bias (which is shown to be common).


4) The methods need more details on the size of the area the plants were collected from. If it was large, the environmental variation among sites in the field could have left its signature on the communities in the individual plants. The plants were grown for 4 months in the lab before analysis. This might have helped reduce environmental influences on the community. However, the environment that they came from will still have possibly defined the pool of fungi that were in the roots, unless there was dispersal into the roots during the 4 months in the lab.

On the other hand. The 4 months in the lab will be a major filter on who is seen as more active vs less active (RNA vs DNA) in the fungal community, making it difficult to understand what might be driving differences in phylogenetic structure among plants in this controlled environment. Do the plants differ in their traits due to genetic differences? Were there differences in age among the plants? This issues is not discussed but should be a major discussion point.


5) A general comment about the logic of line 94, concerning how the endophytic life style could be phylogenetically conserved.
Different environmental conditions may favor or limit transcription for different fungal species or lineages that are all within one functional group. For example, it is well established that some mycorrhizal taxa are common in some ecological conditions (e.g., post fire, etc) relative to others that favor different conditions (e.g., non-disturbed, etc). In the specific ecological conditions that your samples were collected in, it would be expected that certain fungi are favored but under different conditions (e.g., add some nitrogen), other taxa would be favored, even though all are within the same functional group. I don't think you can make broad statements about the "endophytic lifesytle" because it encompasses fungi with extremely wide diversity of functions (e.g., latent pathogens, weak parasites, mycorrhizal fungi, dark septate endophytes), and species between and within these groups will vary in how much they respond (indicated by how much they are transcribing) to different conditions. DNA vs RNA will only tell you who is there vs who seems to be more active out of the pool of taxa that are there.


6) Need more clarify on the calculation of the mean expression ratio. In the methods, it is unclear if the ratio is calculated for each individual OTU in a sample, or if it is a ratio averaging together info from all OTUs in a sample, or if it is about each individual OTU that averages info across all samples. By the time the reader get's to the discussion, I think it become implicit that the mean expression ratio if calculated on a per-plant basis, but the reader should have this solidified in their mind in the methods section.


7) Line 291-297: A described here, it makes me think that DNA are RNA communities were combined when generating null distributions (e.g, reporting 20 out of 38 fractions being significant). If so, 20 is about half the fractions and this result could simply be due to differences between DNA and RNA communities. My interpretation here may not be correct but the passage at least needs to be clarified. I don't think DNA and RNA should be combined in creating null distributions unless a comparison between both is part of the question being address in that specific analysis.

Validity of the findings

1) See notes about the issues of taxonomic resolution and the genomic region that was sequenced, which I made under "Experimental Design". This point follows naturally from that point. In several places the authors make inferences about "species" (e.g., lines 337, 404, 453) but their data does not provide information on species. it is much coarser in resolution. The results may change drastically between different levels of taxonomic resolution. Please make inferences at the appropriate level.

2) See point 4 under "experimental design". I am having a difficult time understanding what could be driving the differences in phylogenetic structure among plant individuals. What is it about the plants? If there are no clear differences, I feel like the results would be due to microsite environmental difference in the field, which would provide that bounds on what fungi are in the roots when they were moved to the lab.


3) This point is relevant to methods and interpretation The paper shows interesting patterns but there is a lack of basic mycological interpretation. Who are the taxa driving the patterns (at a finer-scale than just the phyla)? For example, are there different glomeromycota families that seem to co-exist in some situation than others and what are their general ecological functions/traits. There is lots of information in the literature on fungal functional traits, but the lack of taxonomic information provided on what/who is driving patterns makes it difficult to interpret results with any applicability to understanding the details of nature. The results are significant and interesting, but are very abstract because they are not strongly related to the biology of the lineages in the communities. One way to get more info on who is in the communities is to use BLAST in NCBI to get a better idea of who they are. Or, use phylogenetics to relate the sequences to known clades of fungi. This would be very useful.


4) In general, a lot of points from the "experimental design" section could also be made here, because the methods affect interpretation.

Additional comments

Please read my notes on the attached pdf.

Annotated reviews are not available for download in order to protect the identity of reviewers who chose to remain anonymous.

Reviewer 4 ·

Basic reporting

The article follows the PeerJ policies and is clearly written. An accession number for the sequencing project is provided. Although the authors provide a link to the dedicated analysis pipeline, the pipeline is not publicly accessible and it is thus difficult to assess in depth the data analysis.

Experimental design

The research question is defined at the end of the introduction. In general, the applied methodology and technical standards are sound. The main issue is related to replication (see below).

Validity of the findings

My main concern about the manuscript is related to the experimental design. A main focus in the analysis is the identification of the core fungal microbiota members. However with the applied experimental design, this is not feasible and this issue should be addressed in a revised version. The authors only include one biological replicate using one very specific soil type (turf/peat) in their design. I would expect that a broad range of soil types, genotypes and environmental conditions are tested before defining the core microbiota of Agrostis stolonifera. This is certainly possible since A. stolonifera has a broad ecological niche. Based on the applied set-up, the authors can only infer the between-plant variation (stochastic effcts) in one very specific condition (turf).

In general the sequencing depth is too shallow. This hampers the identification of the core OTUs since many identified OTUs at the RNA level are not detected at the DNA level. Since the overlap between both levels is used to identify the core OTUs, this analysis is biased. In principle, at sufficient sequencing depth, all OTUs should be detected at the DNA level. Rarefaction curves could be informative to the reader to assess the saturation.

Line 361-362: I don’t agree with this sentence. A diversity of fungi is associated with non-mycorrhizal host plants (see Glynou et al 2016) and even in the current study Glomeromycota represent a minor fraction (< 4%).

Figure S1: For transparency of the data, I would include all 19 plants as individual stack barplots in this figure.

---

## Round 0.2 · Minor Revisions

As you can see the revised MS was reviewed by two reviewers (both of whom commented on the original submission), and both now consider that it meets the requirements for publication in Peer J.

However, Reviewer 2 has suggested that further minor amendment is still required and I think has been most helpful in specifying these in the review comments and as annotations on the MS pdf.

I agree with the reviewers that you have done a very good job to address the previous substantive criticisms, and thank you for taking on board the points raised in the review process. I also agree with Reviewer 2's further suggestions for amendment, and would ask you to respond to the points raised by this reviewer.

Reviewer 2 ·

Basic reporting

This article is well written and meets PeerJ criteria. A link to raw data has been supplied as has sufficient detail on how to replicate the bioinformatics pipeline.

Please check manuscript for consistencies with respect to formatting p vales and some other equations: sometimes what is written is correctly formatted (p < 0.05) but oft time one or more spaces have been omitted i.e p< 0.05 or p<0.05 or q=0

Line 313-4 “Otherwise, they were not found in the RNA fraction of all samples and thus did not belong to the RNA core.”

This sentence seems redundant and confusing to me. I think it is re-iterating the idea of what is considered a core OTU? But only referring specifically to RNA OTUs/fraction? Possibly this line can be deleted or re-written for clarification.

Other minor suggestions for improvement have been supplied in attached pdf

Experimental design

This research is within the aims and scope of PeerJ. The research question is clearly defined in the introduction as:“what governs fungal-endophyte-community assembly?” and the authors also highlight the benefit of co-extracting RNA and DNA for community analysis. This research question is highly relevant to the field of microbial ecology. I found the experimental methods and design to be robust and have no additional comments.

.

Validity of the findings

The data from this research was examined in a statistically sound fashion. The conclusions drawn during the discussion are appropriate and relevant. However I would make one observation:

The authors note that there is no significant difference in OTU richness between DNA and RNA fractions for Chao1 estimates of OTU richness (which attempts to estimate true richness by accounting for rare OTUs), yet they conclude that there are significant differences in OTU richness based on the species richness estimate from the hill analysis (q=0). The authors spend a significant proportion of the discussion reporting on the significant differences in OTU richness based on the latter measure, but do not mention result of the Chao1 index, which suggests that the differences are not significant. (although they do highlight that other measures of diversity were n.s). Given that a major finding is that DNA and RNA fractions have different OTU richnesses, the chao1 data should be incorporated into the discussion regarding the conclusion that OTU richness is different. Although not significantly different, the chao1 results exhibit the same trend of decreased richness in the RNA.

Additional comments

Other minor suggestions for improvement have been supplied in attached pdf

Annotated reviews are not available for download in order to protect the identity of reviewers who chose to remain anonymous.

Reviewer 4 ·

Basic reporting

See previous review report

Experimental design

no comment

Validity of the findings

no comment

Additional comments

The authors have well addressed the points raised in my review report and in my opinion the manuscript can be published in its current status.

---

## Round 0.3 · accepted · Accept

Many thanks indeed for taking time and care in revising your paper and fully addressing the substantive points raised in the review process. We greatly appreciate your having addressed these matters so thoroughly, and hope you will agree that the paper has been improved as a result.